Stress-regulated elements in Lotus spp., as a possible starting point to understand signalling networks and stress adaptation in legumes

Menéndez Ana B. 1 2 anamen@bg.fcen.uba.ar
Ruiz Oscar Adolfo 3
1 Departamento de Biodiversidad y Biología Experimental. Facultad de Ciencias Exactas y Naturales., Universidad de Buenos Aires , Ciudad Autónoma de Buenos Aires, Overseas , Argentina
2 Instituto de Micología y Botánica, Consejo Nacional de Investigaciones Científicas y Técnicas (CONICET) , Ciudad Autónoma de Buenos Aires, Overseas , Argentina
3 Instituto Tecnológico de Chascomús, Consejo Nacional de Investigaciones Científicas y Técnicas (CONICET) , Chascomús, Buenos Aires , Argentina
Winkler Robert
Electronic publication date: 2021 Nov 30
Publication date: 2021
Volume: 9
Electronic Location ID: e12110
Received 2021 May 13; Accepted 2021 Aug 14
Copyright: © 2021 Menéndez and Ruiz
Copyright year: 2021
Copyright holder: Menéndez and Ruiz
License: This is an open access article distributed under the terms of the Creative Commons Attribution License, which permits unrestricted use, distribution, reproduction and adaptation in any medium and for any purpose provided that it is properly attributed. For attribution, the original author(s), title, publication source (PeerJ) and either DOI or URL of the article must be cited.
License URL: https://creativecommons.org/licenses/by/4.0/

Keywords: Biotic, Abiotic, Transcription factors, Signaling, Protein-protein interaction, Forage, Lotus, Stress, L. japonicus, Stress improvement

Funding: Consejo Nacional de Investigaciones Científicas y Técnicas University of Buenos Aires This work was supported by the Consejo Nacional de Investigaciones Científicas y Técnicas and University of Buenos Aires. The funders had no role in study design, data collection and analysis, decision to publish, or preparation of the manuscript.

==============================
Although legumes are of primary economic importance for human and livestock consumption, the information regarding signalling networks during plant stress response in this group is very scarce. Lotus japonicus is a major experimental model within the Leguminosae family, whereas L. corniculatus and L. tenuis are frequent components of natural and agricultural ecosystems worldwide. These species display differences in their perception and response to diverse stresses, even at the genotype level, whereby they have been used in many studies aimed at achieving a better understanding of the plant stress-response mechanisms. However, we are far from the identification of key components of their stress-response signalling network, a previous step for implementing transgenic and editing tools to develop legume stress-resilient genotypes, with higher crop yield and quality. In this review we scope a body of literature, highlighting what is currently known on the stress-regulated signalling elements so far reported in Lotus spp. Our work includes a comprehensive review of transcription factors chaperones, redox signals and proteins of unknown function. In addition, we revised strigolactones and genes regulating phytochelatins and hormone metabolism, due to their involvement as intermediates in several physiological signalling networks. This work was intended for a broad readership in the fields of physiology, metabolism, plant nutrition, genetics and signal transduction. Our results suggest that Lotus species provide a valuable information platform for the study of specific protein-protein (PPI) interactions, as a starting point to unravel signalling networks underlying plant acclimatation to bacterial and abiotic stressors in legumes. Furthermore, some Lotus species may be a source of genes whose regulation improves stress tolerance and growth when introduced ectopically in other plant species.

Introduction

Impoverished agricultural soils and problems derived from climate change generate abiotic and biotic stresses that diminish plant growth and crop productivity (Boyer, 1982; Guttikonda et al., 2014). Therefore, the implementation of transgenic and editing tools to develop stress-resilient genotypes, with higher crop yield and quality, has turned into a major biotechnological target. For this purpose, a previous identification of candidate genes, metabolites, and mechanisms controlling plant adaptation to these stresses is required. Mechanisms of plant adaptation include intricate regulation networks relying on the appropriate perception, distribution, integration and processing of the stress signal, all of which is mediated by physical protein-protein interactions (PPI). Some of these proteins are highly connected and play a central role in modular organization of PPI networks (Jeong et al., 2001; Han et al., 2004; Albert, 2005; He & Zhang, 2006; Dietz, 2008; Zhang, Gao & Yuan, 2010; Vandereyken et al., 2018). These proteins are of critical importance as their elimination or interfering with, leads to drastic changes in the structure of the biological network, having a vast impact on the organismal fitness (He & Zhang, 2006). Therefore, their identification and characterization constitute a relevant starting point for the development of crop varieties, which are better adapted to abiotic factors. In fact, large-scale plant PPI networks and interactome studies have been experimentally reconstructed in several plant species (reviewed by Vandereyken et al., 2018), with network elements being detected in the model Arabidopsis thaliana (Mukhtar et al., 2011; Lumba et al., 2014; Trigg et al., 2017) and in important crops such as Oryza sativa (Seo et al., 2011) and Solanum lycopersicum (Yue et al., 2016). Although legumes are of primary economic importance for human and livestock consumption (Duranti & Gius, 1997; Graham & Vance, 2003), little is known about interactions among stress-responsive proteins in the Leguminosae family. Studies in this area are restricted to one performed on a non-model, yet under-exploited legume species (drought stressed Macrotyloma uniflorum; Bhardwaj et al., 2016). Lotus japonicus is regarded as a major experimental legume model. This species displays characteristics that are useful for the generation of diverse genomic tools and biotechnological resources such as http://www.kazusa.or.jp, (Sato & Tabata, 2006; Sato et al., 2008) and the co-expression Toolkit (CORx) at https://lotus.au.dk/ (Mun et al., 2016), turning L. japonicus a suitable platform for legume crop improvement. In addition, several Lotus species display high adaptability to diverse abiotic stresses, making them important components of grassland ecosystems in environmentally constrained areas, where these species are used for livestock production, dunes revegetation, or reclamation of contaminated soils (Escaray et al., 2012). Although some progress has been made to achieve a better understanding of the plant stress-response mechanisms in Lotus, we are far from implementing transgenic and editing tools to develop stress-resilient genotypes, with higher crop yield and quality in these species. To attain this major biotechnological target, a previous identification of key components of the plant stress-response network is required.

Major classes of plant stress response-related signalling elements are transcription factors (TFs), kinase, phosphatases, ubiquitin system associates, chaperones, co-chaperones, redox signals (summarised by Vandereyken et al., 2018). The involvement of these elements in the signalling network may be mapped in silico, but then it needs to be validated using reliable experimental approaches like yeast two-hybrid (Y2H), bimolecular fluorescence complementation (BiFC), or affinity purification and mass spectrometry (AP-MS) assays. In this review we scope a body of literature, highlighting what is currently known on the stress-regulated signalling elements so far reported in Lotus spp., which presented PPI interacting properties by datamining and/or experimentally (themselves or their homologs in other plant species). Phytohormones were included because of their importance in signalling crosstalks. Stress-related signalling processes in the specific root nodule environment were left aside in the present review for being worthy of a separate review, and because several comprehensive reviews, or studies dealing with regulatory aspects of the nodule stress metabolism just came out (i.e., Giovannetti et al., 2019; Pérez-Delgado et al., 2020; Signorelli et al., 2020; Sharma et al., 2020; Hidalgo-Castellanos et al., 2021). Our purpose is to scope a body of literature, which could be either a helpful precursor to systematic reviews of signalling networks studies in Leguminosae, or a starting point to identify potential metabolic hubs. This work was intended for a broad readership in the fields of physiology, metabolism, plant nutrition, genetics and signal transduction.

Survey methodology

We surveyed in Google Scholar and Scopus, the literature relevant to the key words “Lotus (excluding Nelumbo nucifera)”, “L. japonicus”, “L. corniculatus”, “L. tenuis”, “L. glaber” or “legume”. These keywords were used in different combinations with “interactome”, “transcriptomic”, “metabolomics”, “proteomics”, “stress”, “biotic”, “protein-protein”, “transcription factor”, “hormone”, “signal transduction” or “signalling”, with no time interval. The search was expanded to the names of first and senior authors of referent publications.

Transcription factors (TFs)

TFs proteins bind to DNA-regulatory sequences to modulate the rate of gene transcription. Therefore, their functional analysis, including their interaction with other molecules during biotic or abiotic stress is a key step to understand the signalling cascades that lead to plant adaptation.

According to the L. japonicus gene annotation from the Kazusa (v3.0) database, over 2050 TFs, classified into 56 families have been identified in this species (PlantRegMap/PlantTFDB v5.0 Plant Transcription Factor Database Previous version: v3.0 v4.0; http://planttfdb.gao-lab.org/index.php?sp=Lja). Here we summarized those TFs that were reported to be regulated by stress in Lotus species (Table 1; Fig. 1).

Figure 1 Signalling and regulators elements overview.

Overview of signalling and regulators elements detected or regulated during Lotus species responses to different abiotic and biotic stresses. Elements include transcription factors (grey ingots), phytohormones (grey barrels), redox signals (paving stone), antioxidant enzymes (ovals), co-activators (green ingots) and miscellaneous (turquoise ingots). Metabolites relevant for plant defence are also depicted (grey ovals). Arrows and simple lines mean, respectively, the inductions/reactions, and physical interactions described in the text, with those in non-Lotus species depicted with dashed lines. For elements abbreviations refer to text.

Table 1 Overview of studies refering Lotus species and stress-regulated signaling elements.

Authors	Lotus species	Gene or metabolite	Stress*	
Transcription factors	
Babuin et al. (2014)	L. japonicus	bHLH	NaHCO3 (alkalinity)	
MYB	
Bordenave et al. (2013)	L. japonicus	bHLH	Pseudomonas syringae	
MYB	
AP2/ERF	
WRKY	
Calzadilla et al., (2016a)	L. japonicus	DREB1CBF	Cold	
NAC	
AP2/ERF	
MYB	
WRKY	
Escaray et al. (2017)	L. corniculatus, L. tenuis	MYB, bHLH	Ns	
Ke et al. (2018)	L. japonicus	WRKY/C2H2	Salinity	
Kunihiro et al. (2017)	L. japonicus	MYB	Ns	
Lin et al. (2014)	L. japonicus	Hsf	Multiple stresses	
Ling et al. (2020)	L. japonicus	TCP	Drought/Salinity	
Paolocci et al. (2005)	L. corniculatus	bHLH	Ns	
Paolocci et al. (2011)	L. corniculatus	MYB	Ns	
Shelton et al. (2012)	L. japonicus	MYB/bHLH	Ns	
Soares-Cavalcanti et al. (2012)	L. japonicus	Hsf	Multiple stresses	
Sun et al. (2014)	L. japonicus	ERF	Salinity	
Sun et al. (2016)	L. japonicus	AP2/ERF	Salinity	
Yoshida et al. (2008), Yoshida et al. (2010)	L. japonicus	MYB	Multiple stresses	
Redox signals, antioxidants and compatible solutes				
Bordenave et al. (2017)	L. japonicus	Lactic, threonic, succinic and p-coumaric acids; valine and β-alanine	Alkalinity	
Calzadilla et al. (2016b)	L. japonicus	Glutathione transferase	Cold	
Trx-TrxR	
Melchiorre et al. (2009)	L. japonicus	SOD, GR, APX	Salinity	
L. filicaulis	
L. burtii	
Matamoros et al. (2020)	L. japonicus	GSNOR	Ns	
Ramos et al. (2007)	L. japonicus	PCS	Heavy metals	
Rocha et al. (2010)	L. japonicus	Alanine, succinate	Waterlogging	
Rubio et al. (2009)	L. japonicus	CAT, DR, MR	Salinity	
Shimoda et al. (2005)	L. japonicus	Hbs	Cold	
Signorelli et al. (2019)	L. japonicus	NO	Drought	
Sanchez et al. (2012)	L. japonicus, L. japonicus, L. filicaulis, L. burttii, L. corniculatus, L. tenuis	proline	Drought	
Sanchez et al. (2008)		amino acids	Salinity	
L. japonicus	sugars	Salinity	
	polyols	Salinity	
Phytohormones				
Bordenave et al. (2013)	L. japonicus	JA	Biótico (Pseudomonas syringae)	
ET (ACC synthase, ACC oxidase)	
SA (NPR3; EDS5)	
ABA	
Auxines	
Babuin et al. (2014)	L. japonicus	Gibberellins	Alkalinity	
Auxines	
JA	
Espasandin et al. (2014)	L. tenuis	ADC	Drought	
Menéndez et al. (2019) (Review)		Polyamines	Multiple stresses	
Pandey, Sharma & Pandey (2016)	L. japonicus	Strigolactones	Drought	
Tapia et al. (2013)	L. japonicus	JA	Drought	
Miscellaneous				
Calzadilla et al. (2019)	L. japonicus	DnaK, GroEL (chaperon/chaperonin)	Cold	
Kojima et al. (2013)	L. japonicus	ARM	Drought/Salinity	
Notes:

Ns, no stress factor analyzed in the study.

* Experimental or based on published scientific reports.

Basic/helix-loop-helix (bHLH) form the largest TFs family in L. japonicus and comprise two distinct functional regions, a basic N-terminal DNA-binding region, and a helix-loop-helix region involved in PPI that functions as a dimerization domain (Murre, McCaw & Baltimore, 1989; Feller et al., 2011). Several probesets assignable to bHLH TFs were strongly regulated in a microarray-based analysis of the transcriptomic L. japonicus responses to the inoculation with Pseudomonas syringae (Bordenave et al., 2013). During the bacterial pathogenesis, most bHLH genes were repressed in the sensitive genotype (MG-20), whereas two up-regulated, bHLH-proteins were observed among the genes identified in the tolerant genotype (Gifu B-129). Later, the global transcriptomic responses of the same L. japonicus accessions were analysed on plants exposed to long-term alkaline stress (Babuin et al., 2014). Again, various bHLH-like probe sets were greatly induced in the tolerant genotype (MG-20), whereas the majority of bHLH-like genes resulted down-regulated or slightly induced in the sensitive (Gifu B-129) one. These results are interesting as several network forming bHLH proteins, with relevant functions during plant response to biotic and abiotic stress were detected in other plant species by data mining and experimentally. For example, in rice and chickpea (Singh et al., 2015), and Arabidopsis (Van Moerkercke et al., 2019), MYCs (bHLH TFs) interact in vitro and in vivo with proteins involved in several jasmonic acid signalling pathways.

MYB (v-myb avian myeloblastosis viral oncogene homolog) constitute another large family of TFs In L. japonicus. MYBs represent a family of proteins that include the conserved MYB DNA-binding domain (Stracke, Werber & Weisshaar, 2001), in plants characterised by the R2R3-type MYB domain. Three copies of the TRANSPARENT TESTA2 (TT2, a homolog of an Arabidopsis MYB), were found in L. japonicus by Yoshida et al. (2008). Their results from yeast two-hybrid experiments showed that LjTT2a interacts with TT8 and TTG1 (encoding bHLH and WDR TFs proteins, respectively), whereas its expression in plant was related to environmental stress tolerance, and to the accumulation of proanthocyanidins (PA, flavonoid end-products, Robbins et al., 2003). Furthermore, through expression analysis (Yoshida et al., 2010), it was demonstrated that the ectopic, combined expression of MYB and bHLH (and WDR) regulated one promoter member of the dihydroflavonol 4-reductase (DFR2), the first committed enzyme of the flavonoid pathway, which leads to common anthocyanins (another flavonoid end-products) and PA biosynthesis. Similar MYB and bHLH co-modulations at the flavonoid pathways have also been shown for other plant species such as maize (Goff, Cone & Chandler, 1992) and Arabidopsis (Nesi et al., 2001). These works give support to the idea that proteins of the MYB and bHLH TF families can form MYB/bHLH complexes to regulate distinct cellular processes or metabolic pathways, including those taking part of plant response to stresses (Pireyre & Burow, 2015). In L. japonicus in fact, a reduced expression of a MYB-bHLH complex would permit the derivation of metabolites from the flavonoid pathway to the isoflavonoid biosynthesis (Shelton et al., 2012). On other hand, the overexpression of LjMYB12 (ortholog of an Arabidopsis MYB TF) in L. japonicus resulted in the upregulation of genes coding for the chalcone synthase paralog CHS1 (although not of other paralogs), a key enzyme of the flavonoid/isoflavonoid biosynthesis pathway (Kunihiro et al., 2017). In addition, several MYB TFs transcripts were found among genes showing maximal regulation during the response of two L. japonicus ecotypes (MG-20 and Gifu B-129) to alkaline and biotic stresses by Babuin et al. (2014) and Bordenave et al. (2013), respectively. Interestingly, a lower proportion of up-regulated, or even repressed Myb-like genes were detected in the corresponding tolerant L. japonicus genotype in each stress. All the above information becomes relevant as diverse roles for many isoflavonoids in plant defence against biotic stresses, as well as in the acclimatation to abiotic stress have been pointed out (Agati et al., 2012). Based on above mentioned reports, it could be hypothesized that the observed divergences in the expression patterns of MYB-like and bHLH genes, between sensitive and tolerant L. japonicus ecotypes might lead to dissimilar arrangements of flavonoids and isoflavonoids, which could have accounted for different stress tolerances.

MYB transcription factor physically interacts as well with the promoters of CBF/DREB (C-repeat binding factor/dehydration-responsive element binding), major elements regulated by low temperature and water deficit, (Stockinger, Gilmour & Thomashow, 1997). In L. japonicus, cold stress-induced changes in the expression profile of L. japonicus DREB1/CBF genes were reported, in congruence with MYB TFs being found among genes showing maximal upregulation during plant response to cold (Calzadilla et al., 2016a). Last observed outcomes were congruent with a previous report in Arabidopsis, where MYB15 negatively regulated the expression of the CBFs, whereas its overexpression resulted in decreased tolerance to freezing stress, and its knock-out mutant exhibited increased Arabidopsis freezing tolerance (Agarwal et al., 2006).

Taken together, former results advocate for a role of L. japonicus MYBs and bHLH interacting proteins in the signalling network that leads to stress responses. They also invite to perform specific PPI or in silico studies to gain insight into MYBs and bHLH topological roles as “hubs” or “bottlenecks” (number of interactions that these proteins may hold within those networks; Dietz, Jacquot & Harris, 2010).

In parallel, stress induced modulation of MYBs and bHLH TFs levels in some Lotus species may have severe implications on livestock breeding. L. corniculatus and L. tenuis are herbaceous forage legumes highly valued by livestock producers (Sato & Tabata, 2011). L. corniculatus leaves contain high levels of PA (also known as condensed tannins), polymeric flavonoids which are present in legumes used as forage of high nutritional value, while PA are absent in L. tenuis leaves. As PA prevent bloating in ruminant animals, obtaining plants of forage legumes with engineered PA traits constitutes a pertinent strategy to increase sustainability of cattle production systems, from both the ecological and economical perspectives. To identify among candidate PA regulators and transporters, the expression analysis of genes related with PA biosynthetic pathways was studied in interspecific hybrids between L. corniculatus and L. tenuis, and their F2 progeny (Escaray et al., 2017). As result, it was found that leaf PA levels significantly correlated with the expression of: MATE1 (the putative glycosylated PA monomers transporter), MYB genes TT2a and TT2b, MYB14 and the bHLH gene TT8 (authors considered the last four as candidate regulatory genes). In addition, the expression levels of TT2b and TT8 also correlated with those of all key structural genes of the PA pathways investigated, including MATE1.

L. corniculatus was also used to test the effect of the ectopic expression of TFs genes on flavonoids accumulation and plant stress tolerance. The transgenic over-expression of the maize bHLH gene Sn in L. corniculatus, led to subtle anthocyanin accumulation, while PA were dramatically enhanced in the leaf blade, without altering other major secondary end-products such as flavonols, lignins and inducible phytoalexins (Robbins et al., 2003). Through a real-time RT-PCR approach, Paolocci et al. (2005) showed that the same exogenous bHLH TF affected in a lesser extent the expression of early genes (PAL and CHS), compared to later genes of the pathway (DFR and ANS), whose mRNA levels were increased. In turn, CT accumulation derived from the increment of DFR and ANS genes was limited by light. In addition, authors discussed that, to obtain the desired effect, trans-activation should be targeted to the specific iso-forms members of such gene families. Also, the flavonoid biosynthesis repressor from strawberry FaMYB1, tissue-specifically suppressed PA biosynthesis in L. corniculatus transgenic plants (Paolocci et al., 2011). More recently, the overexpression of an ERF gene cloned from Kandelia candel (KcERF), along with a DREB TF cloned from Populus euphratica (PeDREB2) improved drought and salt tolerance in transgenic L. corniculatus (Wang et al., 2018). Former information could be relevant as L. corniculatus is one of the most important forage legumes with high nutritive value worldwide, and breeding L. corniculatus with enhanced stress tolerance would be an interesting biotechnological goal. In addition, it constitutes an approach for controlling flavonoids biosynthesis given that leguminous plants use them during their interactions with other organisms, and in response to various environmental stresses (Aoki, Akashi & Ayabe, 2000).

APETALA2/ETHYLENE RESPONSIVE FACTOR (AP2/ERF) proteins, one of the largest TFs families in L. japonicus are defined by the AP2/ERF domain. These TFs regulate a variety of biological processes related to growth, development, and responses to environmental cues, as they are involved in DNA binding (Nakano et al., 2006). In Arabidopsis, AP2/ERF genes play a hub role during plant survival to stress conditions, by integrating regulatory networks during specific stress responses, and by helping to activate ethylene (ET) and abscisic acid (ABA) dependent and independent stress-responsive genes (Xie et al., 2019; Müller & Munné-Bosch, 2015). In L. corniculatus cultivar Leo, the transcription of LcERF genes was strongly induced by salt and stress-related phytohormones (Sun et al., 2014). The overexpression in Arabidopsis of LcAP2/ERF107, encoding an AP2/ERF protein resulted in enhanced tolerance to salt stress and increased seed germination (Sun et al., 2016), indicating that this gene plays an important role in the responses of plant to salt stress. On other hand, the transcriptomic study performed on L. japonicus plants challenged with P. syringae (Bordenave et al., 2013) revealed different expression patterns in over 15 ERFs genes between two ecotypes (Gifu B-129 and MG20), which are contrasting in their tolerance to bacterial infection. Particularly, two Arabidopsis ERF3 and ERF4 homologs were found to be repressed in the sensitive L. japonicus ecotype (MG-20). In addition, an ERF5-like factor was also repressed in the sensitive one. These results deserve a deep insight in the future, as the overexpression of ERF genes enhances plant resistance to pathogens in different plant species (Xu et al., 2011; Moffat et al., 2012).

C2H2 zinc finger proteins (ZFP) form a relatively large family of transcriptional regulators in L. japonicus. ZFPs emerge as possible hubs, as they interact with other zinc finger proteins or other protein types to regulate target gene expression (Brayer & Segal, 2008; Song et al., 2010). For example, direct interactions between ZAT6, an Arabidopsis C2H2-type ZFP, and other proteins (LDL, MPK and TPL) were found on basis to experimental evidence (BioGRID database; https://thebiogrid.org/15592/table/arabidopsis-thaliana/zat6.html). In addition, key roles for C2H2 zinc finger proteins during plant responses to abiotic stresses have been described (Wang et al., 2019; Han et al., 2020). In transgenic L. japonicus plants, salt stress induced the overexpression of a GmWRKY protein containing a C2H2 zinc finger motif, leading to increased salt tolerance, compared with the wild type (Ke et al., 2018). Combined, these reports turn worthwhile to search for the possible interactions between C2H2 and other proteins taking part during L. japonicus response to salinity.

NAC TFs is a family of proteins sharing a highly conserved N-terminal DNA-binding domain and a variable C-terminal domain (Ooka et al., 2003; Fang et al., 2008, 2020). NACs are highly represented among annotated genes in the Kasuza L. japonicus TFs database. Up-stream regulators of NAC genes and down-stream NAC target genes have been reviewed by Jensen & Skriver (2014). Last authors revealed NAC molecular interactions, signal pathways intersections and biological functions with relevance for agriculture. Furthermore, NACs show potential as candidates to produce plants with enhanced multiple stress tolerance (reviewed by Shao, Wang & Tang, 2015). In Lotus, information on the NAC involvement in stress response is limited to a transcriptomic study performed on L. japonicus plants confronted with cold stress, where NACs were detected among the most numerous up-regulated TFs, along with AP2/ERF, MYB and WRKY families (Calzadilla et al., 2016a).

WRKY TFs proteins function as repressors and de-repressors of important plant processes, including responses to biotic and abiotic stressors (Song et al., 2018). The common feature of WRKY proteins is the presence of an approximately 60-amino-acid DNA-binding domain, known as the WRKY domain, followed by a zinc-finger motif at the C-terminus (Chen et al., 2019). In Arabidopsis and rice, members of the WRKY TF family were shown to regulate immune responses by reducing their susceptibility to pathogens (Asai et al., 2002; Abbruscato et al., 2012).

Transcriptomic allowed the identification of 10 upregulated and two down-regulated WRKY-like genes upon Pseudomonas syringae inoculation in a sensitive L. japonicus, whereas the four most regulated of these transcripts were also up-regulated in the tolerant genotype (Bordenave et al., 2013). Later, a whole L. japonicus genome analysis (Song et al., 2014), allowed the detection and analysis of 61 putative WRKY genes which were classified into three groups (LjWRKY I–III). To study the role of these WRKY TFs in the mechanism of tolerance against P. syringae further research is needed, including cloning, sequencing of coding protein, and PPI analysis such as Y2H studies. It is worth to note that caution is needed at transferring outcoming results to the biotechnology field as divergent effects were found among WRKY proteins, and their over-expression may enhance stress susceptibility (Kesarwani, Yoo & Dong, 2007).

Heat stress transcription factors (Hsf) and proliferating cell factor (TCP) genes are among the less represented, stress-involved TFs in the L. japonicus database. Hsf are major regulators leading to the activation of genes responsive to heat stress, and many chemical stressors (Kotak et al., 2004). The Hfs repertoire in L. japonicus was identified and characterized, resulting in 19 candidate ESTs for this species (Soares-Cavalcanti et al., 2012). Later, the evolution of Hsf genes in six legume species, including L. japonicus, was studied by Lin et al. (2014). These authors showed the whole genome duplication origin for the vast majority of Hsf gene duplications. The putative involvement of L. japonicus Hsf genes in numerous tissues or developmental stages, and in response to several abiotic stresses was also revealed by the same authors using expression analysis. On other hand, a total of 25 proliferating cell factor (TCP) genes were identified from the genome of L. japonicus in a genome-wide analysis (Ling et al., 2020). The promoter analysis revealed that cis-elements were related to stress responses, in line with results of qRT-PCR indicating that the L. japonicus TCP genes played regulatory roles in both salt and drought stresses.

Redox signals, antioxidants and compatible solutes

Drought, salinity and cold may induce the overproduction of reactive oxygen species (ROS) and nitric oxide (NO), which may contribute to the negative effect of stress caused by the oxidative damage at the cell level (reviewed by Noctor, Reichheld & Foyer, 2018). However, endogenous levels of some of these molecules may also play a role in cell signalling, allowing a fast response to metabolic changes. Notwithstanding this, signalling reactions need to be tightly controlled to maintain the cellular redox balance and prevent damage.

NO regulates root architecture through cross talks with other gaseous molecules like hydrogen sulfide (H2S) and carbon monoxide (CO), often in association with several other growth regulators, whereby deciphering these interactions could be a potential biotechnological tool to improve crop production, particularly under restrictive soil environments (Mukherjee & Corpas, 2020).

NO is a crucial molecule indirectly regulating signalling cascades that may affect protein functions in different L. japonicus organs (Matamoros et al., 2020). Under drought stress, L. japonicus displays an active NO metabolism characterized by an increase of protein nitration and NO (and S-nitrosothiols) accumulation in root cortical cells (Signorelli et al., 2019).

NO concentration may be modulated by nonsymbiotic hemoglobins (Hbs) in all plant organs, by binding it to the heme group, or to cysteine thiol groups (Igamberdiev & Hill, 2004; Bustos-Sanmamed et al., 2011). It has been shown that cold induces LjHb1 (expressed in all plant tissues; Shimoda et al., 2005), in line with the cold-driven induction observed on another type of Hb gene in Arabidosis leaves (AtGLB2, Trevaskis et al., 1997). Interestingly, the overexpression of LjHb1 in L. japonicus plants decreased NO levels and delayed nodule senescence (Fukudome et al., 2019). These results suggest that nonsymbiotic Hbs could modulate the signalling cascades triggered by NO during plant adaptation to cold stress, thus improving plant growth. In addition, the functional and structural characterization of two nonsymbiotic Hbs in L. japonicus could facilitate the transfer of genetic and biochemical information of these relevant proteins into crops (Calvo-Begueria et al., 2017).

NO may react with glutathione (GSH) to form S-nitrosoglutathione (GSNO; Kovacs & Lindermayr, 2013). GSNO participates in protein S-nitrosylation, an ubiquitous posttranslational modification wherein nitric oxide (NO) is covalently attached to a thiol group of a protein cysteine residue (S—NO; Stamler & Hess, 2010), which impacts on cellular signalling and plant response to abiotic and biotic stresses (Corpas, Alché & Barroso, 2013; Kubienová et al., 2014). In turn, the S-nitrosoglutathione reductase (GSNOR) descomposes the S—NO bonds in Arabidopsis (Sakamoto, Ueda & Morikawa, 2002). Recently, two GSNOR genes were identified and characterized in LjGSNOR1 and LjGSNOR2 mutants by Matamoros et al. (2020). The Ljgsnor1 mutant contains 19 proteins that are specifically S-nitrosylated and are involved in defence and stress responses (besides protein degradation, hormone biosynthesis and photosynthesis). Authors showed that the activity of LjGSNOR1 and LjGSNOR2 proteins was sharply affected by H2O2 and H2S, leading them to suggest that GSNORs may be important regulatory hubs by integrating signals mediated by H2O2, NO and H2S.

Major components of the plant antioxidant battery in plants include ascorbate, glutathione and the enzymes superoxide dismutase, catalase, glutathione peroxidase and the four enzymes of the ascorbate-glutathione cycle: ascorbate peroxidase, monodehydroascorbate reductase, dehydroascorbate reductase and glutathione reductase (Nakano & Asada, 1981; Dalton et al., 1986; Pignocchi & Foyer, 2003). The impact of salt stress on the levels of most of these enzymes was analysed as part of the antioxidant defence response characterization in model and non-model Lotus species (Melchiorre et al., 2009; Rubio et al., 2009). The antioxidant response to combined drought-heat (Sainz et al., 2010) and to low temperature (Calzadilla et al., 2016b) were also analysed on L. japonicus. Plants of this species exposed to chilling revealed the up-regulation of a glutathione transferase transcript, which is compatible with H2O2 and other ROS accumulation (Calzadilla et al., 2016b).

Thioredoxin is a class of small proteins playing many important biological processes, including redox signalling. In plants, it is well known that the Trx-TrxR system (thioredoxin, peroxiredoxin and thioredoxin reductase) takes part in ROS detoxification, protein redox regulation, and various signalling mechanisms (Foyer & Shigeoka, 2011). In a study addressing the photosynthetic acclimation response of two Lotus japonicus ecotypes (MG-1 and MG-20), with contrasting tolerance to cold stress, the better photosynthetic performance of the tolerant ecotype (MG-20) could be due to the higher Trx-TrxR protein levels found in that ecotype, what would have led to a higher ROS detoxification capacity (Calzadilla et al., 2019).

Phytochelatins (PCs) are Cys-rich, enzymatically synthesized peptides playing an essential role in heavy metals detoxification in numerous Phyla, including plants (Pal & Rai, 2010; Hasan et al., 2017). In L. japonicus, the expression of PCs genes was examined (Ramos et al., 2007) and three functional LjPCS genes were identified and found to be differentially expressed in roots during response to Cd. However, the overexpression of phytochelatin synthase gene AtPCS1 in Arabidopsis did not enhance tolerance to heavy metal stress (Lee et al., 2003). It has been stated that the failure of breeding programs where gene variation at the structural or expression level does not result in the predicted stress tolerant phenotype, may be due to the involvement of metabolites, multigenes and post-translational modifications, not identifiable by genomics or transcriptomics approaches (Mazzucotelli et al., 2008; Weckwerth, 2011).

Proteomics and metabolomics approaches provide information on levels of anti-stress components such as compatible solutes, antioxidants, and stress-responsive proteins, and on metabolic reprogramming associated with stress tolerance (Wienkoop et al., 2008; Kosová et al., 2011; Obata & Fernie, 2012; Doerfler et al., 2014). Using gas chromatography/mass spectrometry-based allowed the detection of changes in amino acids, sugars, and polyols profiles of L. japonicus plants subjected to long-term salinity (Sanchez et al., 2008; Fig. 2), upon salt stress. The same technique was used to study the changes that occur in the global primary metabolome profile of L. japonicus MG-20 (tolerant) and Gifu B-129 (sensitive) when treated with 30 mM NaHCO3 (Bordenave et al., 2017). The study revealed differential accumulations of lactic, threonic, succinic and p-coumaric acids, as well as valine and β-alanine between both ecotypes. The last amino acid (along with succinate) also accumulated in L. japonicus plants subjected to waterlogging-induced hypoxia, due to the activation of alanine metabolism, and the splitting of tricarboxylic acid pathway (Rocha et al., 2010).

Figure 2 Stress-induced metabolites in Lotus japonicus.

Scheme of most induced primary metabolites in Lotus japonicus plants exposed to drought, alkalinity and water logging.

A comparative metabolomic study indicated a relative low degree of conservation among metabolic responses to drought among L. japonicus, L. japonicus, L. filicaulis, L. burttii, L. corniculatus, L. tenuis (ex L. glaber) and L. uliginosus (Sanchez et al., 2012). Despite this, proline accumulated in all genotypes except L. uliginosus. Proline is one of the best studied stress-related amino acid (Rana, Ram & Nehra, 2017) and its physiological role has been highlighted among the global metabolic rearrangements in drought-stressed L. japonicus plants (Diaz et al., 2010).

Phytohormones

Phytohormones are key regulators of multiple developmental processes (Bunsick et al., 2021) and take part of many signalling networks that control stress responses (Ku et al., 2018). Because phytohormones are natural and non-toxic compounds, it was suggested that their application as chemical control agents could be safe and environmentally friendly (Wang et al., 2020).

Combined molecular, biochemical, and genetic data have shown that jasmonic acid (JA) is a key signalling molecule during the plant defence response to pathogens and abiotic stress (Ruan et al., 2019; Wang et al., 2020). In L. japonicus, methyl-JA contributes with plant acclimation to drought stress by promoting cuticle synthesis through the up-regulation of LjLTP6, a gene involved in cutin formation (Tapia et al., 2013). Results from a global transcriptomic study performed by Bordenave et al. (2013) revealed contrasting results between L. japonicus ecotypes MG-20 and Gifu B-129, regarding their JA response to Pseudomonas syringae infiltration. Induction of the JA synthesis pathways was observed in the bacteria-sensitive L. japonicus ecotype MG-20, but not in the tolerant Gifu B-129 one (Bordenave et al., 2013). The activation of the JA pathway in MG-20, could eventually have implications on plant nitrogen nutrition of MG-20, as it was shown that shoot-derived JA could function as a negative regulator on nodulation (Nakagawa & Kawaguchi, 2006). In fact, JA is considered a signal intervening in the photomorphogenetically controlled nodulation of L. japonicus MG-20 by the red/far red (R/FR) ratio (Suzuki et al., 2011). The above mentioned transcriptomic studied also revealed the up-regulation in MG-20 leaves, of genes coding for aminocyclopropane-1-carboxylic acid synthase and 1-aminocyclopropane-1-carboxylic acid oxidase, enzymes implicated in the ET biosynthesis, supporting that ET often works synergistically with JA during pathogen attack (Glazebrook, 2005).

It is known that the activation of the salicylic acid (SA) pathway in plants is associated with the production of several pathogenesis related (PR) proteins, which play diverse defence roles (Loake & Grant, 2007). In Arabidopsis, the expression of PR genes mediated by the SA pathway requires the transcriptional co-regulators NPR1 and Enhanced disease susceptibility 1 (EDS1; Loake & Grant, 2007). Interestingly, NPR3 and EDS5-like homologs were found downregulated in MG-20 upon the infection, but no regulation of these genes was found in Gifu B-129 (Bordenave et al., 2013). Also, most of the genes related to the Abscisic acid (ABA) metabolism were negatively regulated after MG-20 leaf infiltration with P. syringae. This result was congruent with the occurrence of dehydration symptoms surrounding the infiltration point, as ABA plays a crucial role in the adaptation to water stress (Nakashima & Yamaguchi-Shinozaki, 2013). In contrast, ABA-related genes remained unregulated in Gifu B-129. In parallel, transcripts of GH3-like genes (which code for IAA-amido synthetases involved in auxins regulation; Bari & Jones, 2009) were up-regulated in both ecotypes. Authors concluded that differences in the levels of activation/repression of genes related to the regulated hormonal pathways between both L. japonicus ecotypes could explain their different sensitivity to bacterial infection. A similar experimental setup allowed the same research group to register the differential expression, between MG-20 (tolerant) and Gifu B-129 (sensitive) ecotypes, of several hormone-related probe sets, upon alkalinization with NaHCO3 (Babuin et al., 2014). Auxin-responsive genes, as well as genes involved in gibberellin and JA biosynthesis were induced in both ecotypes, mostly in shoots, whereas a negative regulator of the gibberellin signal transduction pathway, a RGA-like probe set (cm0584.17_at: Silverstone, Ciampaglio & Sun, 1998) was detected in MG-20 roots. In contrast, probe sets putatively related to ABA and ET biosynthesis were induced, and genes probably involved in auxin and cytokinin response or biosynthesis were down-regulated in Gifu B-129 roots.

Polyamines (Pas) are natural aliphatic amines involved in many physiological processes, including responses to abiotic stresses, in almost all living organisms. It has been postulated that molecules generated from their catabolism may act as secondary messengers, taking part of signalling networks in numerous developmental and stress adaptation processes (Moschou, Paschalidis & Roubelakis-Angelakis, 2008). Piled up evidence in L. japonicus and L. tenuis, that links variations in Pas titles and stress tolerance could foster the identification of stress tolerant phenotypes and promote their yield and adaptation to constraint environments by the sketching of new biotechnological approaches (Menéndez et al., 2019). For example, in L. tenuis, the overexpression of ADC2 (a key enzymes involved in the biosynthetic pathway of putrescine, one common polyamine and precursor of other Pas) driven by the stress-inducible RD29A promoter, improved drought tolerance in plants subjected to a gradual decrease in water availability (Espasandin et al., 2014).

Strigolactones (SLs) bear a carotenoid structure, and function as plant growth and development regulators, modelling the plant architecture in response to biotic and abiotic stimuli. The structurally simplest known SL is 5-deoxystrigol (5-DS), which was isolated from L. japonicus root (Pandey, Sharma & Pandey, 2016). In this species, osmotic stress alters transcription of genes encoding for SL biosynthesis and transport, decreasing SL levels. On the other hand, the osmotic stress-induced ABA (also a carotenoid phytohormone) accumulation was inhibited in L. japonicus plants treated with SL, as it down-regulates the ABA biosynthetic gene LjNCED2. These results suggest that osmotic stress promotes precursors derivation from the SLs biosynthetic pathway to the ABA biosynthetic pathway, confirming a SL-ABA cross talk (Liu et al., 2015).

Miscellaneous: chaperones and flowering-related regulators

Armadillo repeats (ARM) consist of tandem repeats forming a right handed superhelix of α-helices (Huber, Nelson & Weis, 1997), whose function in Arabidopsis are related to protein–protein interaction during abscisic acid signal transduction (Kim et al., 2004) and epidermal-cell morphogenesis (Sakai et al., 2008). The armadillo (ARM) repeat-like protein of L. japonicus LjTDF-5 plays a role in response to high-salt stress and dehydration (Kojima et al., 2013). This gene is homologous of the G. max transcript-derived fragment 5 (GmTDF-5), which has been identified as a salt-inducible gene (Aoki et al., 2005) and it is considered to interact with other protein(s) responsible for abiotic stress response, in soybean. In addition, increased levels of DnaK and GroEL (chaperon and chaperonin, respectively) in response to low temperature were detected to a greater extent in a tolerant L japonicus ecotype (MG-20), compared to the sensitive one (MG-1; Calzadilla et al., 2019). Several stress-related functions have been assigned to these molecules. Importantly, chloroplast chaperonins have been implicated in RuBisCo biosynthesis, suggesting their involvement in the photosynthetic acclimation response (Gruber et al., 2013).

Finally, plant response pathways activated by biotic and abiotic stresses can modify processes under the circadian clock control, including flowering (Kazan & Lyons, 2016; Kugan et al., 2021). Although it is known that drought-stressed Arabidopsis plants accelerate or delay flowering under long or short days, respectively (Riboni et al., 2016), the information regarding the links between changes in clock gene expressions and stress-induced physiological variations in Legumes is extremely scarce (Kugan et al., 2021). The fact that phase modulation was found to be involved as a mechanism to alleviate iron deficiency symptoms in soybean (Li et al., 2019), opens the question for the occurrence of a similar phenomenon during the stress responses in Lotus spp. A starting point to find an answer should be the study of the relatively few signalling elements reported intervening in flowering, and other circadian-controlled processes, which would be worthwhile being evaluated in relation to biotic and abiotic stresses, i.e., the putative PHYTOCHROME INTERACTING FACTOR4 (LjPIF4, Ono et al., 2010) Ljmybr, an MYB-related gene (Duangkhet et al., 2016), clock-associated F-box proteins (Boycheva et al., 2015), NOOT-BOP-COCH-LIKE (NBCL) regulators (Magne et al., 2018) and the flowering regulation gene (Lj2g3v1989150, Shah et al., 2020).

Despite L. japonicus being a long day species, there are intrinsic differences in flowering time among L. japonicus ecotypes due to their latitudinal distribution in origin, as occurs in Japan (Kawaguchi, 2000). For example, the Miyakojima MG-20 ecotype flowers much earlier than the Gifu B-129 ecotype. Therefore, the stress-driven delay in flowering should be considered when these L. japonicus ecotypes are compared. Likewise, the flowering time may not be trivial when it comes to biomass production of forage species like L. corniculatus or L. tenuis, as the later the plant blooms, the greater is the biomass accumulation.

Conclusion and perspectives

The genus Lotus contains several species and genotypes differing in their perception and response to diverse stresses, which constitutes a valuable platform to study and understand common and specific PPIs underlying plant acclimation to bacterial infection and abiotic stressors (drought, salts, cold and heavy metals). Major pathways involved in transcriptional and metabolic reprogramming to improve plant survival of Lotus spp. plants include those related with flavonoids and phytohormones. Some Lotus species may be a source of genes whose overexpression improves stress tolerance and growth, even when they are introduced ectopically in other plant species. Also, Lotus species proved useful for testing the effect of foreign gene expression on stress tolerance improvement.

Most of the genes or molecules here reviewed, which could theoretically function as signalling or physiological hubs, have been studied in the model L. japonicus (Table 1), with TFs as the best represented group. To gain insight into the involvement of these elements in the plant response regulatory network, it would be worthwhile performing clustering analyses of differentially expressed genes (DEGs) using the Lotus japonicus expression atlas (https://lotus.au.dk/expat/), which would allow to find groups of genes with similar gene expression patterns. This would complement global in silico PPI network analysis, using web tools such as the STRING v11.0 (https://string-db.org/) or BioGRID4.4 (https://thebiogrid.org/) servers. Since no PPI database is available for L. japonicus, in silico analysis of PPI networks should be performed by using their homologs in other legumes (e.g., Medicago truncatula). This strategy allowed the identification of nitrate reductase as a central gene for the regulation of nodule function in L. japonicus (Pérez-Delgado et al., 2020). We consider that conducting in silico analysis of PPI networks in the responses of Lotus species to stress constitutes a very relevant, exciting and probably fruitful objective in itself, which deserves to be addressed in a specific research work, exclusively for this purpose. Once detected, potential hubs must be confirmed through experimental yeast 2-hybrid procedures, or in planta screenings. Further, to study the structural basis of these interactions may reveal intersections among pathways involved in more than one stress response. The integration of genomics, proteomics and metabolomics, with computational predictions and experimental PPI identifications techniques, would allow a detailed understanding of molecular networks underlying plant stress response, a prerequisite for engineering of important agronomic traits to be applied in crop breeding programs.

Additional Information and Declarations

Competing Interests

Author Contributions

Data Availability

The authors declare that they have no competing interests.

Ana B. Menéndez conceived and designed the experiments, performed the experiments, analyzed the data, prepared figures and/or tables, authored or reviewed drafts of the paper, and approved the final draft.

Oscar Adolfo Ruiz conceived and designed the experiments, authored or reviewed drafts of the paper, discussion and critical review of the manuscript, and approved the final draft.

The following information was supplied regarding data availability:

This is a literature review.

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
