# Peer review of "Stress-regulated elements in Lotus spp., as a possible starting point to understand signalling networks and stress adaptation in legumes"

_PeerJ, doi:10.7717/peerj.12110_

## Round 0.1 · original submission · Major Revisions

Both reviewers suggest changes in the structure and additional contents. As well, the final manuscript should be carefully checked for language issues.

Reviewer 1 ·

Basic reporting

The authors have submitted a review on stress-regulated elements involved in stress adaptation signaling in Lotus spp. This is a very interesting and missing piece of information in legumes. A list of interesting examples of diverging stress-related behaviors among the Lotus species are reviewed.

Experimental design

The survey methodology should be improved (see below)

Validity of the findings

The reported information are important to postulate interplay between different biotic and abiotic stresses and responsive elements, paving the way for the obtainment of legume stress resilient genotypes. The review is well written and the reading flows nicely.
See also general comments below

Additional comments

I have two major criticisms.
The L. japonicus MG20 and Gifu are the two accessions mostly used as model systems for fundamental studies. The authors report the Sato et al. 2006 reference (web site: http://www.kazusa.or.jp/lotus/index.html) where information and tools are provided. I found quite strange that all over the review, the authors never cite the Mun et al. 2016 paper (web site: https://lotus.au.dk/) , where some innovative tools are also provided. In particular, the co-expression toolkit is something that the authors should have exploited to support (if possible) some of the interesting correlations they report, aimed to identify putative protein-protein interactions playing roles in the stress-related signaling pathways.
Another missing point is the lack of information on the pathways affecting the flowering time in Lotus spp, the TF involved and the correlation with stress conditions. This is a crucial phenotypic treat for biomass production in forage legumes and the MG20 accession, which is known as an early flowering accession, could provide important clues.

Minor points
I would suggest to divide the two main sections, Transcription Factors and Chaperons, redox….. in subtitled paragraphs. The subtitles of the TF section could be the names of the different TF families analyzed. In the case of the second section, I would suggest to subtitle with the different kind of stresses analyzed (e.g. Salt stress, drought…).
On my view, the paragraph on the CBS motifs (lines 294-302) could be eliminated. There are too poor information reported in literature to state their roles in stress response pathways.
Line 108, In vitro in vivo. Italic

Reviewer 2 ·

Basic reporting

This contribution is a review of bibliography of the proteins and genes involved in stress signaling in several Lotus spp (as expected, mainly from the model L. japonicus). The focus is on stress signaling networks and the paper is organized in two major blocks: "transcription factors" and a miscellaneous section including "chaperones, redox signals and proteins of unknown function". There are abundant studies on transcription factors and more scattered information on the other types of proteins. I would make "hormones" a separate section, as they cannot be included in the second block. The authors briefly describe polyamines, strigolactones, ethylene and ABA. The description of results on these hormones needs to be expanded and should encompass other hormones and defense-related compounds such as jasmonic acid and salicylic acid. For clarity and easier reading, the authors should consider also making a specific block for antioxidants, which can include information on thioredoxins and phytochelatins from the second block, as well as reports on ascorbate metabolism, thiol metabolism, superoxide dismutases and catalases. mention to reactive oxygen and nitrogen species (RONS) is also lacking. These reactive species are clearly involved in stress signaling. There are several papers linking these enzymes and metabolites in the response to drought and salt stress. English is quite good and the text can be read without problems, but the paper needs to be revised by a native speaker because there are numerous errors of syntax and grammar. A double-check for correct citations in the text and literature secion needs also to be conducted (for instance, citation of Xie et al. 2019 is not included in the bibliography section). Also, the paper requires a thorough revision of English language.

Experimental design

The article is within the scope of the journal. The survey of literature is unbiased but incomplete. Studies on antioxidant defenses in response to stress and on additional hormones and growth regulators are lacking. Also, in most cases it is unclear if the data come from non-nodulated plants or from nodulated plants. That is important because nodulation may make a major difference in sensitivity/tolerance to abiotic stress. Blocks of information (sections) need to be reorganized. Also, there seems to be little connection between the sections. It would be desirable to try to put the review of genes/ proteins into the context of a metabolic and regulatory framework. In this respect, one or two figures addressing the pathways of stress defense and signaling would facilitate reading and would make the review more appealing.

Validity of the findings

The novelty of the literature survey is justified as it includes both the model Lotus japonicus and other Lotus spp. (L. tenuis and L. corniculatus) that are relevant as cultivated plants or in natural ecosystems. The conclusions that L. japonicus can be used as a model to study stress signaling is straightforward. However, a similar study could have been performed using another model legume such as Medicago truncatula and its related crop species M. sativa. In other words, it is fine to focus on Lotus but similar conclusions could have been obtained by examining literature of molecular stress responses in Medicago. The review is of interest to a broad readership, ranging from plant physiologists and agronomists to geneticists and molecular biologists.

Additional comments

The review is a literature survey of Lotus spp. in response to stress. The idea and timing for this review are fine, but the paper needs reorganization, expansion and some "glue" between sections. One or preferably two figures are required to provide a global picture of the genes/protein being reported. This would also enhance the interest and understanding of the review. Correct correspondence of citations between text and bibliography section needs to be revised. English needs also to be heeded.

---

## Round 0.2 · accepted · Accept

The structure and the flow of the manuscript have been improved. Your review about stress-regulated elements in Lotus spp is a valuable contribution to plant research.

Reviewer 1 ·

Basic reporting

The authors have properly responded to all the comments and submitted an improved version of the review. I'm fully satisfied with the revised manuscript.

Experimental design

The study design has been improved

Validity of the findings

The review summarizes an important issue in legumes

Additional comments

The authors have properly responded to all the comments and submitted an improved version of the review.